# Fostering Hope: Comprehensive Accessible Mother-Infant Dyad Care for Neonatal Abstinence (CAIN)

**DOI:** 10.3390/children9101517

**Published:** 2022-10-04

**Authors:** Denise Clarke, Karen Foss, Natasha Lifeso, Matthew Hicks

**Affiliations:** 1Northern Alberta Neonatal Intensive Care Program, Stollery Children’s Hospital, Alberta Health Services, Edmonton, AB T6G 2B7, Canada; 2Neonatal Intensive Care, Women’s & Child Health Program, Covenant Health, Edmonton, AB T6L 5X8, Canada; 3Faculty of Medicine, University of Calgary, Calgary, AB T2N 4N1, Canada; 4Faculty of Medicine and Dentistry, University of Alberta, Edmonton, AB T6G 1H9, Canada

**Keywords:** hope, neonatal abstinence syndrome, neonatal opioid withdrawal syndrome, mother-infant, grounded theory, qualitative research

## Abstract

Hospital and community healthcare providers have expressed concerns around the continuity and quality of care for infants with neonatal abstinence syndrome (NAS) during hospitalization and transition home. This qualitative study explored the experiences of hospital and community-based healthcare providers and identified themes related to the management of NAS for mothers and infants. Healthcare providers that cared for women with substance use disorders and/or cared for newborns with NAS in a large urban setting in Canada met inclusion criteria for this study and were interviewed in groups or as individuals. Interview transcripts were reviewed iteratively using inductive thematic analysis to identify an overarching theme linked with primary themes. In total, 45 healthcare providers were interviewed. Qualitative analysis of their experiences derived the overarching theme of hope with five primary themes being: mother/infant, mental health, system, judgement, and knowledge. The study identified gaps in NAS care including fear, stigma, and language. This research demonstrates that programs and interventions that work with mothers and newborns with NAS must foster hope in mothers, families, and in the extended care provider team and improve communication between hospital and community networks.

## 1. Introduction

Neonatal abstinence syndrome (NAS) occurs in a significant proportion of newborns exposed to substances in utero. More recently, the term neonatal opioid withdrawal syndrome (NOWS) has been introduced, but the term NAS is still used for cases of polysubstance exposure [1]. In Canada, maternal opiate use in pregnancy has been steadily rising resulting in as many as 1850 newborns born with NAS per year [2,3,4]. Management of newborns with NAS requires supportive, interdisciplinary care [5]. Depending on the severity of NAS, newborns may be cared for with nonpharmacological interventions in the same room as the mother or be admitted to the Neonatal Intensive Care Unit (NICU) to receive medication. The Canadian Pediatric Society’s (CPS) position statement supports a more holistic approach to the management of NAS by advocating for rooming-in care for parents and families, placing less emphasis on pharmacotherapy, promoting breastfeeding, increasing and supporting parental presence, and highlighting the need to keep the family together [2]. The Fir Square Combined Unit at BC Women’s Hospital and Health Centre in Vancouver has used the rooming-in model of care as the standard practice for opioid-dependent women and their infants for the past 10 years with several recently published articles attesting to the safety of this approach [6,7,8,9,10].

Newborns experiencing NAS, especially those requiring medication treatment, often have extended hospital stays and complex ongoing care needs [2,11]. Not only is there an increasing trend in keeping the postpartum mother-baby dyad together thereby decreasing NICU admissions, but also a move towards supportive earlier outpatient care for those infants requiring medication to manage NAS. Comprehensive discharge planning and outpatient support for infants with NAS aims to balance safety with achieving goals to decrease length of hospital stay and overall hospital costs. [12,13,14,15,16,17] A systematic review showed that cautious awareness is required with this cost-driven management approach: families with infants with NAS may have poor attendance with outpatient appointments and are at risk of being lost to follow up. [18] As well, infants with NAS have higher risk of emergency department visits, hospital readmission, and early childhood mortality [18].

Hospital and community healthcare providers have expressed concerns around the continuity of care for these newborns in their stay in hospital and transition from hospital to home. These concerns highlight the need for intensive supportive management for the overall health and well-being of the mother-baby dyad both in and out of hospital, yet there is a paucity of research addressing the perspectives of healthcare providers who interact with these families and their infants with NAS during these critical transition points. The purpose of this study was to gain an understanding of the experiences of hospital and community healthcare providers regarding the management of newborns at risk for NAS and identify key themes and potential areas for improvement with hospital-based NAS care and subsequent transition to community care.

## 2. Materials and Methods

### 2.1. Study Design

This interpretive inquiry study used a constructivist grounded theory approach [19,20] to explore themes from guided individual and focus group interviews of individuals with experience with NAS care in a large city in Canada. In general, grounded theory is a methodology used to construct an explanatory model about psychosocial experiences of interest [21]. There are three grounded theory approaches, all of which utilize inductive procedures to enable researchers to understand the phenomenon being studied using constant comparison and simultaneous data collection [22]. However, the three approaches differ in how prior knowledge and researcher experience is viewed. In the first approach, classical grounded theory, no previous literature review is done prior to data collection because the theory should be derived solely from the data collected, not from a preconceived ideas about the data [22,23,24]. The second postpositivist approach shifts to accepting the possibility of multiple viewpoints including the influence of the researcher’s subjectivity [21,22]. Constructivist grounded theory is the third approach. While all grounded theory approaches encourage an in-depth review of literature over time as theory emerges [19,20,21], the constructivist orientation differs somewhat in that it recognizes the usefulness of a preliminary review of existing literature, allowing the researcher to be aware of the broader context that impacts study participants’ experiences [23,24]. The constructivist grounded theory approach also expands upon the postpositivist viewpoint expressly acknowledging that researchers cannot maintain complete objectivity and incorporates steps to minimize subjectivity [19,20,22,23]. In other words, this approach considers previous personal and professional experiences of the researchers [22]. For example, most of the researchers in our group were not coming to this research as blank slates. We have extensive clinical experience with NAS and we know many of the research participants in a professional capacity. Any personal biases this prior knowledge and experience may have created were addressed through debriefing after each participant encounter, as well as methodologic rigor and trustworthiness further discussed in Section 2.4.

### 2.2. Participants

Targeted recruitment from healthcare professionals working in women and childrens health in the hospital setting and community programs occurred. This was done through posters and an information email sent to nine in-patient units at three hospitals as well as community partners with the mean length of each interview being 45 min in duration. All self-identified recruits were invited by email to participate in the focus group interviews. Interested healthcare providers were invited to participate in focus groups and individual interviews. The hospital personnel were Registered Nurses, Neonatologists and Social Workers from NICU, Labor and Delivery, and Postpartum. Community personnel included Registered Nurses, Peer Support workers, Social Workers, and Managers from programs such as Public Health, Children Family Services, as well as specific programs focused on addiction recovery and the needs of marginalized populations. In part, the focus group conversations helped to identify key individuals that were subsequently invited to participate in individual interviews. Those interviewed alone included a community Pharmacist, a Postpartum Manager, a community Nursing Director, a Neonatologist and a Pediatrician. These individuals work closely with families experiencing NAS and expressed a keen interest in participating in this study. Although the study team was prepared to expand recruitment of healthcare providers if necessary, reviewers identified distinct recurring patterns in the data, with no new themes emerging before the interviews concluded. Thus, no further recruitment was undertaken.

### 2.3. Data Collection

A semi-structured interview guide was used to facilitate dialogue with participants about the challenges and experiences of providing NAS care in hospital, community, and home settings. Interview guide development was informed by team member extensive clinical experience with NAS care and processes and practices identified in a separate pilot project related to nonpharmacological management of NAS [25,26] (Table 1). The semi-structured guide was reviewed by experienced clinicians and management team members prior to being used. The guide was revised based on team observations of responses in early focus groups. Focus group interviews were conducted at three hospitals and one community facility, while individual interviews occurred at each participant’s location of choice. Informed consent was obtained from each participant and interviews were audio-recorded then transcribed verbatim. Participants also had the option of providing demographic data on a standardized data collection sheet.

### 2.4. Qualitative Analysis

Qualitative analysis of interview transcripts occurred over several months and was completed in December 2019. This analysis was facilitated by MindNode (version 2.5.8) (IdeasOnCanvas GmbH, Vienna, Austria) to identify themes associated with the dynamics of caring for infants at risk for NAS and their families. Early in the analysis process, a small panel of NICU team members, including a social worker, registered nurse, and NICU family mentor volunteer, was consulted for their initial impression of the data, and to independently confirm that saturation had been reached indicating that no further interviews were needed. The main researchers (DC, KF, MH, NL) met several times over a 15-month period to conduct ongoing conceptual analysis supported by a more detailed exploration of each transcribed interview with MindNode to identify main themes, subthemes and coded quotes associated with the central concept of NAS. MindNode provides an approach like mind mapping. An advantage of mind mapping is that it allows free thinking and diverse perspectives, displaying participant-centric linkages and facilitating comparisons in a visual way that might not otherwise have been evident, while attempting to manage the large volume of raw data common with qualitative methods [27,28,29,30,31]. This thematic analysis culminated with the identification of key categories in a connected nodal structure that formed the basis of the study’s findings. Methodologic rigor and trustworthiness, as per constructivist grounded theory, was ensured through considering credibility, originality, resonance, and usefulness at each step of the work [20]. Credibility was ensured through interviewing a diverse sample group of care providers, as well as through peer debriefing [32] at two separate time periods in the analysis process. Further, member checking to establish credibility and get feedback on our data interpretation [32] occurred by way of an online meeting to share the nodal structure and summary of findings with a small group of study participants. Originality, resonance and usefulness came by way of using the final nodal structure to present data in a unique and meaningful way that not only supports interpretations derived from what participants have expressed, but also creates the underpinnings of a theory that is applicable in a broader context such that it could inform care of infants with NAS and their families beyond our local experiences.

## 3. Results

Study interviews were conducted from March to June 2018 with five individual and four focus group interviews conducted. In total, a purposive sample of 45 individuals drawn from hospitals and community health organizations were interviewed. Most participants elected to not provide the optional demographic data. Care providers were encouraged to draw from their experiences interacting with newborns with NAS and their families. The overarching theme identified in our analysis was of hope. Further to this overarching theme, five primary themes were identified: mother/infant, mental health, system, judgement, and knowledge (Figure 1).

Hope emerged in all participant groups as a central and unifying theme. Hospital personnel expressed hope for the mother–infant dyad to remain intact, that the family continued safely on their journey, and that there was long term, sustainable success. Community personnel expressed hope that hospital staff would achieve a better understanding of community resources, trauma informed care, and that successful hospital interactions with their patients would produce an intact mother–infant dyad. Hospital staff also expressed that an absence, or death, of hope was related to previous negative experiences and struggles in connecting families with adequate resources and supports. Healthcare providers identified that they felt that they provided suboptimal care related to an absence of hope and this contributed to judgement and stigma of families.

From our analysis, participants also described five dominant sub themes to the primary theme of hope. The sub themes included the following: mother/infant, mental health, system, judgement, and knowledge. Components of these themes were identified as facilitators (Table 2) or barriers (Table 3). Barriers were the obstacles staff and families encountered; whereas facilitators were the supports/complements.

### 3.1. Mother/Infant

Intact mother–infant dyads were identified as one of the primary themes in our study. The overall goal is to always maintain the mother/infant dyad and the integrity of the family. This relationship is essential when managing NAS as it is well known separating mother and newborn can have detrimental effects on early attachment, mother’s mental health, as well as the newborn’s neurodevelopmental outcome. Pursuing this goal provides hope for families and staff. A subtheme to this was the mother’s resiliency, also identified as a facilitator. “[These mothers] were addicts but they were incredibly resilient and could survive what many of us could not … they are hunting and gathering every day”. Two other facilitators were supporting the mother’s well-being and involving mothers and families in the care of the newborn. Treating mothers and the families with “acceptance and no judgement” is key and identifying that “this is the only mother this child will ever have” and realizing that “this isn’t just about the baby, that there is a mother involved and a family” were highlighted themes. The Canadian Pediatric Society’s position statement supports a more holistic approach to the management of NAS by advocating a model of rooming-in care for parents and families, placing less emphasis on pharmacotherapy, and highlighting the need to keep the family together. From our data, healthcare professionals identified the importance of keeping mother–infant dyads intact and providing this hope for mothers.

### 3.2. Mental Health

Mental health was another primary theme identified with the mental health concerns common secondary to intergenerational trauma and complex social situations. If the mother’s mental health is supported, she will feel hopeful that she will be able to care for her child. A facilitator identified for mental health was to break the cycle for the mother. Providing her with safe, consistent support is essential for the mother to be able to care for her newborn “to fill that hole in their soul, so at least we have planted a seed; they’ve remembered one person who was kind and maybe they will come sooner, or they will access healthcare a little bit better”. An obstacle to achieving stable mental health was substance use disorder (SUD). If the mother does not have a support person or conducive environment, stable mental health is challenging. Substance use disorder is complex and complete abstinence may not be achievable, but support and treatment programs are. “I think we all just need to see the mom as a real key to making this work … if she feels welcome … she will show up and she will be there, she won’t leave”. Supporting the mother and encouraging her will provide her the hope and strength to persevere.

### 3.3. System

The primary theme of system included community, public health, and hospital settings. All pieces of the system must work together to provide the means for the mother and infant to stay together for a positive long-term outcome. It is ideal to engage with the mother in the antepartum period and follow her throughout the continuum of her pregnancy to the postnatal period. For this to occur and be successful, all three components of the healthcare system must work in unison. Subthemes identified were all facilitators and they included: creating continuity of care, accessibility to care, streamlined communication, and focused care. The goal is to create “a community service that they can connect with during the pregnancy … the biggest thing is a consistent person … a support person”. Once in hospital, it was identified as essential to have consistent caregivers involved, “there is social worker involvement and a pediatrician lined up and we have steps in place for the baby to go home”. Streamlined communication from hospital to community was identified as important to ensure follow up. “You almost need an abstinence team … you know a team that looks after these newborns … that work together with these families”. Addressing accessibility to care to ensure the family attends follow up appointments was another important factor. “[Supporting] the family to break the cycle of trauma” and offering them nonjudgemental care when they come in for their appointments is needed. Participants identified that many of these mothers have endured years of trauma and judgement and do not trust the system. It is imperative that the care provided to them is supportive, consistent and trauma informed. “Trauma informed care … leads to harm reduction and better outcomes”. Seamless, accessible, supportive care along the pregnancy spectrum provides hope for the mothers and the healthcare personnel invested in these families.

### 3.4. Judgement

The primary theme of judgement was also identified along with several subthemes. The sub themes identified as barriers were the following: language, stigma, stereotypes, and effects on relationships. “There is a lot of stigma attached to the term [NAS] and associated with the Mom”. There can be stereotypes placed on these mothers and families and they are treated differently by some of the healthcare staff. “How are you going to get better if everybody takes your baby away and you are treated like you are useless”. They are also often spoken to differently and at times with condescension. The judgement they endure deters them from caring for their newborn and this has a lasting detrimental effect on their long-term attachment. These identified barriers cause them to lose hope, and without hope they lose the ability to connect with their newborn. A facilitator to the system’s primary theme was the self-awareness of healthcare staff. “I think we all just need to see the mom as a real key to making this work … if she feels welcome and she feels like she is an important part of making baby healthy, she won’t leave”. If non-judgemental care can be achieved, this provides hope for mothers and hope for the staff that the family may be able to remain intact.

### 3.5. Knowledge

Possessing the right knowledge is necessary to properly care for the mothers and their newborns at risk of NAS. Staff who desired information were facilitators for this theme. The staff that sought to learn more about social circumstances and choose to utilize this knowledge to better support and encourage these mothers were identified as facilitators. “Staff education [is key] … learning about … helping with that early withdrawal … early feeding, skin to skin”. Providing medical care along with social care is essential for success, “[supporting] the parents, so that they actually feel they can be here, they can bond with their baby, and they can feel like mom and dad”. Fear was identified as an obstacle for this theme. Staff felt fear for the safety of the newborn, “it’s for good intentions that you are caring for that baby and protecting that baby, but sometimes that care and protection actually pushes the parents away”. Staff education on NAS, trauma-informed care, and harm reduction were identified as components to facilitate hope for these mothers and promote mother–infant bonding and attachment.

## 4. Discussion

This study set out to gain an understanding of experiences pertaining to the management of NAS newborns. Hope emerged as the overarching theme across all participant groups. Following Snyder [33], we define hope as individuals having the capacity to find ways to achieve goals coupled with having the motivation to sustain them over time. According to the hope model conceptualized by Snyder and colleagues [33,34,35], there are three interrelated basic mental components of hope. In this model Snyder discusses goals, agency and pathways. Goals are objects, experiences, or outcomes that people imagine and desire [33]. These goals may be abstract or concrete, short-term or long-term, and are the result of mental processes aimed at determining desired outcomes [33]. Goals must be important enough to motivate people, yet they need to have some degree of uncertainty [33]. Agency is the impetus or energy behind hopeful thinking and is otherwise thought of as the willpower or motivational component that propels people along their imagined pathways towards their goals [35]. Willpower relies on people’s perception that they may initiate and sustain actions directed at desired goals [33]. Pathways refer to the perceived ability to produce mental action plans or roadmaps that people envision as being the routes to take towards achieving their goals [34,35]. Pathways thoughts, or waypower, assumes that people have the capacity to engage in planful thought. This capacity is based in part on previous history of successfully finding pathways to goals even when other routes have been blocked [33].

All people have the capacity to hope [36]. Simplistically, hope may be viewed as synonymous with goal attainment [33]. As such, “[hope] is a type of goal directed thinking in which [individuals] perceive themselves as being capable of producing routes to desired goals, along with motivations to initiate and sustain usage of these routes” [34] (p. 25). Having high hope is beneficial for health-related problems, such as through increased ways of thinking about strategies for coping with illnesses (pathways) and through increased ability to use and understand information to make choices about potentially adaptive strategies (agency) [33,34,35,36,37]. Thinking about desired goals has been shown to elicit thoughts about how to attain them [37]. Upon thinking about goals, people imagine being able to work toward and ways to reach their goals [37]. Helping pregnant women with SUD clearly articulate their goals may help them towards both their agentic and pathway thinking. For example, pathways not only encompass what mothers could do to reach their goals, but also what care providers may do to support these women, such as accessibility of care, focused management, consistency and continuity of guidance, and communication.

Recent data show that the odds of opioid use during pregnancy for mothers reporting depressive or anxiety disorders are almost double the odds for women without these diagnoses [38,39]. In the context of these co-occurring mental health challenges, it is important to help pregnant women with SUD realize that important goals take time to reach, and that life throws obstacles in all our paths. Therefore, rather than letting these women assume that they alone are experiencing difficulties, instead help them to normalize these feelings and think about obstacles and failures as challenges [35]. As well, reinforce that it is normal to be psychologically down occasionally. These “psychological troughs” [33] (p. 229) are experienced by high hope people as well but for shorter periods. One way to help them overcome obstacles is to rehearse them by imagining or visualizing oneself in the various stages leading to the goal, anticipating any challenges and planning how to respond to them. Another way to overcome challenges is to encourage these women to ask for help, such as from friends and community agencies [40]. Reinforce that there are instances in all our lives where we cannot come up with ways to achieve our goals in isolation. Friendly exchanges not only support or console someone during difficult times, but also help them strategize about how to reach positive goals, including reinforcement that people often do have some of the skills necessary to problem solve [34]. Hope is both internalized and fostered by those in one’s community, such as peers, family members, service providers, and others supportive of one’s recovery [40,41,42]. Pregnant women with SUD need support but “bonds with sober friends and relatives are weak or broken and the women depend on professional help, which is both needed and feared” [40] (p.465). Hope is an important component of recovery from substance use given that it requires both the motivation to overcome repeated challenges and the ability to generate pathways to overcome these challenges. Interventions must offer these mothers a safe base from which they can explore and reprocess their often-troubled relational experiences, including those with substances [40]. The relational nature of hope highlighted by our study is further supported by findings from a cross-sectional investigation in sober living homes in the United States [42]. In this investigation involving 229 participants it was found that having hopefulness and a sense of community, whereby individuals are connected by common purpose and interpersonal relations, was a predictor of improved quality of life and maintaining successful recovery.

The results of our study highlight three goals of mother and the healthcare system: keeping mother and infant together; supporting mother’s well-being; and breaking the cycle. Development of hope begins early in life [34]. This does not mean that people that do not develop goal setting skills as children cannot learn to acquire these skills in adulthood [34]. Behaviours of those around pregnant women with SUD have a positive or negative impact on development of these goal-related capabilities. For example, health and community care providers may facilitate acquisition of skills through modeling, such as helping these mothers to learn the ‘if-then’ sequence of events that leads to outcomes [33,34,36]. This goal-directed support helps them to begin to develop a plan for change and ways to achieve goals by breaking them into doable, small steps. Even so, work with combat veterans demonstrates that for people who have spent a lifetime trying to forget the past, the idea of intentionally accessing the pain and sharing it with another is frightening [37]. For survivors of trauma, reformulating the goal of forgetting the past into learning how to cope with it is a starting point [37]. Further, helping pregnant women with SUD set self-challenging, yet doable realistic goals begin with generating a list of several desirable aims, prioritizing them, and making those of the highest priority specific and achievable.

Our study demonstrates that hope not only impacts the mother and family, but also the attitudes of health and community care providers. Care of pregnant women with SUD may be difficult and challenging as these relationships are often compounded by lack of trust from both parties [43]. Healthcare participants often expressed feeling cautiously optimistic for the future of the mother–infant dyads, while a small number expressed feelings of hopelessness. We observed that many participants were challenged to answer the interview question about rewards they experienced when working with infants with NAS and their families. There are a few points to consider when developing goal-directed hopeful relationships with women with SUD. First, in terms of relationships, people use components of hopeful thinking when they interact with and try to understand others [33]. Therefore, hopeful thinking is an interpersonal asset and mutual hope is important in interactions between these women and health providers [33]. Second, it is useful to remember that failures occur in strategy, not the people themselves, thus avoiding judgement can help redirect attention back to the goal [37]. Third, it is important to be clear that the pregnant person is establishing a goal because it is something she wants and not a societal “should” [33], such as that all women who become pregnant want to be a mother [40,43]. In an interpretive phenomenological analysis of focus group interviews with a total of 14 pregnant women with SUD, Söderström [40] found that pregnancy is a preparation for parenthood, yet not all women experience this news as positive. Acceptance of pregnancy was seen as a process involving mental representations of self-as-mother that were influenced by past and present experiences as well as by assumptions about the future. Early on, most women had suspicions and sensations of pregnancy, but this situation evoked feelings of guilt and worry, which increased stress, and many continued their substance use. While many of these women had ambivalent feelings about the pregnancy, they were also ambivalent about stopping substance use. Over time, many women experienced mixed feelings with continued ambivalence, vacillating between hope and fear. With emerging hope, the women in Söderström’s study saw pregnancy as a way to survive. Similarly, in our study participants expressed that hope emerged and evolved during pregnancy giving mothers reasons to develop goals, such as taking care of themselves.

The findings of our study also demonstrate that agency impacts the motivation of both mothers with SUD and the healthcare providers they interact with, especially in relation to improving knowledge of care, improving knowledge for support, encouraging resiliency and addressing barriers related to judgement and fear. These findings indicating that increasing hope, particularly agentic thinking, may be beneficial to recovery outcomes are supported in a survey-based study of adults in sober-living recovery programs [44]. Ways for healthcare providers to support agency include: rekindling beliefs about capability to initiate and sustain goal directed action because it is “the rare person whose mental pilot light is totally out” [33] (p.224); creating an internal goal message that is affirmative instead of doubtful; deriving some satisfaction for goals that are achieved; and finding balance between getting there (process) and being there without solely living or being completely controlled by the goal [33]. Söderström [40] found that pregnant women with SUD experience both discouragement and support in their interactions with healthcare professionals. In her study women expressed concerns about their interactions with professionals, viewing their interference as suggesting that they were not fit as mothers, which caused anger and resistance. This made the decision to stop substance use difficult because substances offered comfort and a sense of safety during a time that was experienced as challenging and scary. Partners and grandparents-to-be could offer assurance of help and saw pregnancy as an opportunity for their partner/daughter to become sober, this in turn strengthened the women’s self-confidence and instilled hope for the future [40]. Instilling hope in care providers is equally important to motivate them to further their knowledge of SUD, create desire to support the mother–infant dyad in a non-judgemental manner, and facilitate self-awareness. This instillation of hope includes finding ways to energize and excite these care providers about what they are learning about SUD and discovering about themselves. As well, encouraging health providers to find balance between work which may be emotionally challenging, and personal ways to rest, and recharge is another strategy to support agency within healthcare professionals. This relationship is essential when managing NAS as it is well known that separating mother and infant can have detrimental effects on early attachment. The support and involvement of families in this situation requires a deep commitment by health-care professionals to a nonjudgemental approach to care.

Participants in our study frequently expressed widespread lack of knowledge by health providers as being a barrier to supporting hope in pregnant women with SUD. This lack of knowledge on the part of healthcare providers is pervasive, and impacts attitudes towards mothers of newborns with NAS [45,46]. A suggested first step in overcoming this knowledge gap is helping health providers to understand that there are different degrees and stages of hope: high, low, and the death of hope. It is recognized that the cliche ‘where there is a will there is a way’ is not entirely true because people with willpower thinking (agency) may not have waypower thoughts (pathways) [33,35]. Neither willpower nor waypower alone is sufficient to produce high hope [33]. High-hope thinking provides an advantage when things are difficult because in times of challenge, these individuals think of alternative routes to their goals and apply themselves to the pathway that appears most likely to work [33,47]. In contrast, low-hope individuals experience more negative reactions and are challenged in the face of impediments. Low-hope individuals may focus on a blocked goal, developing an overwhelming sense of futility about how to problem solve [47]. Continued blockages and repeated failures lead them to despair, cynicism and apathy whereby individuals acknowledge defeat and cease all goal pursuits [33,47]. However, as our study shows, this despair and indifference does not have to be an enduring state.

### Limitations

There were several limitations in the study including obtaining a family perspective on their experiences in caring for infants with NAS both in and out of hospital. Future work will attempt to explore themes related to family perspectives in a comprehensive fashion. Another limitation to the study was that professional relationships existed between several interviewers and some personnel interviewed, which may have influenced some responses. Of note, our data are slightly dated, secondary to challenges for the study team in meeting for ongoing progressive thematic analysis given restrictions and clinical demands related to the pandemic. However, the themes identified are robust and unlikely to have changed during this timeframe. Last, there may have been topics that were not covered or discussed that could have potentially provided additional information and/or themes in our conclusions. Although these limitations were recognized, we still achieved saturation for healthcare provider experiences in the themes identified.

## 5. Conclusions

This study expanded upon existing recommendations in the management of newborns with NAS and provided a better understanding of the perceptions and experiences of hospital and community personnel. Despite ever-increasing awareness of concepts such as harm reduction and trauma-informed care, in the neonatal field especially, there often remains a disconnect between the care of women/mothers and their newborns, and consequently a failure to understand that the mother is a large part of NAS management. Our study brings a focus to maternal health and well-being, more specifically coming from a unique lens of hope. This research demonstrates that programs that work with pregnant women with SUD and newborns at risk for NAS must foster hope in mothers, families, and in the extended care provider team.

Hope Theory changes how we think about hope. Whilst hope is an everyday positive sentiment passed from one person to another, it is also a conceptually well-grounded place of opportunity for intentional intervention in pregnant women with SUD and NAS management in newborns. The Hope Theory model discussed includes three mental components: goals (desired outcomes) that are impacted by both agency (motivation to sustain goals over time) and pathways (capacity to find ways to achieve goals). Our study highlights six hope-related subthemes, each contributing to an action plan for NAS management, and more specifically maternal well-being. Hopeful care as it relates to SUD in pregnant women does not occur in isolation, but rather with the help of family, friends, community and healthcare providers. Important first steps of hopeful care include acknowledging the challenges and providing opportunities to deal with history of trauma and co-occurring mental health difficulties to move forward in recovery with a goal of healthy pregnancy and potentially keeping the mother–infant dyad intact should the pregnant person desire that. Things that facilitate hopeful care are the mother’s own resiliency and her social supports, as well as a responsive system that promotes focused management, streamlined communication, and consistency and continuity across the pregnancy continuum, while also addressing barriers to accessible care.

We began this study with expectations of developing an understanding of ways to improve communication between hospital and community, including insights about staff education, follow up care for newborns with NAS, and access to care for the mother–infant dyad. As we reviewed the interview transcripts, employing multiple coding and discussion procedures over several months, our understanding evolved into the idea of hopeful care in relation to NAS management, including an initial look at barriers and facilitators. Additional research would help to ground our findings in what underlying social conditions facilitate hopeful care in comparison to those that inhibit its application to then inform policy recommendations, which is a gap in the current literature. Further, existing research does not provide the perspective of mothers and families about hopeful care. We are currently involved in a pan-Alberta research project which may address these gaps, not just from the perspective of healthcare providers but also by the maternal/family participants.

It is well known that judgement, including language, stigma and stereotypes, is most often a barrier to accessible care and negatively impacts relationships between pregnant women with SUD and health providers. We acknowledge that healthcare providers often come from a good heart and our study shows that this judgement usually stems from acting out of fear for the infant’s well-being. The deleterious impact on relationships, however, may be tempered by self-awareness, information seeking and knowledge utilization on the part of healthcare providers. Even where gaps in knowledge exist, healthcare providers must themselves remain hopeful and provide continued support to break the cycle of hopelessness by helping mothers and families experiencing challenges with NAS believe that they have the necessary skills to achieve their goals, whether that be having a healthy pregnancy, having a positive hospital experience, or taking their newborn home. Knowledgeable facilitative support by care providers is based on an understanding that hope is not a static process, identifying and pursuing goals is not seamless, nor is the continuum of hope linear.

## Figures and Tables

**Figure 1 children-09-01517-f001:**
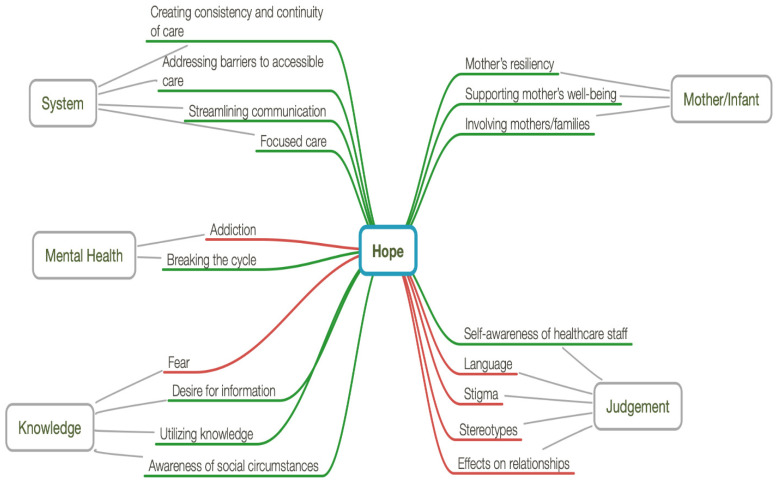
MindNode map identifying themes associated with the dynamics of caring for infants at risk for NAS and their families. The red lines represent barriers to hope within each primary theme. The green lines represent facilitators of hope within each primary theme.

**Table 1 children-09-01517-t001:** Comprehensive Accessible Mother-Infant Dyad Care for Neonatal Abstinence (CAIN) study interview questions.

Category	Example Questions
Knowledge	Tell us what you know about Neonatal Abstinence Syndrome (NAS)/infant withdrawal. What seem to be the gaps in knowledge your co-workers have about NAS?
Experiences (including challenges and rewards)	Please tell us about your experiences with newborns with NAS and their families. What are the challenges working with newborns with NAS and their families? What are the rewards working with newborns with NAS and their families? What are the challenges in discharging newborns with NAS from hospital?
System	What has changed since you first started working with newborns with NAS? What has not changed? What resources are lacking for newborns with NAS and their families? What are the gaps you can think of?
Priority Needs	What do you see as the priority need for newborns with NAS and their families?

**Table 2 children-09-01517-t002:** Facilitators to hope.

Primary Theme	Quotes
Mother/Infant	
Mother’s Resiliency	“I saw that these women were incredibly resilient. They were addicts, but they were incredibly resilient and could survive what many of us could not”.
Supporting Mother’s Well-Being	“I think that goes back again, and this is the second time we are mentioning it, but it is the prenatal piece, so if we can be involved prenatally, as early as possible, and we get to know that mom and she gets to trust us, we can set her up for that success because we can talk to her, we can get a bond and we transition her, so educate her about not just NAS, but about everything; so she can eat healthily, so she can go out for walks every day, so she socializes with other people, invite her to groups. Start to normalize her world with other pregnant women, not just isolated, you know whatever it takes …”.
Involving Mothers/Families	“We all just need to see the mom as a real key part as a real key to making this work”.
Mental Health	
Breaking the Cycle	“… to fill that hole in their soul, um, so at least we have planted a seed; they’ve remembered one person was kind and maybe they will come sooner or maybe they will access health care a little bit better”.
System	
Creating consistency and continuity of care	“… there is a plan going ahead and there is social worker involvement and a pediatrician lined up and we have our steps in place for the baby to go home”.
Addressing barriers to accessible care	“… our clients don’t have an easy time getting to medical appointments, so it’s very hard when they do finally make it there, if they are late they get turned away because they’ve come across the city on the bus with their kids and any number of reasons, um, or when they miss and they are charged fees so therefore they won’t come back, or physicians, family physicians, community physicians won’t see them anymore if missed so many appointments. Those are the families who really need it the most”.
Streamlining communication	“I think it would be safe to say that there is a gap between acute care and community, like we don’t even really know what people mean when they say community, like it’s sort of a magical thing, like we will sort of just catch this baby and this mom and do something with them”.
Focused care	“I think management doesn’t always mean medications, pharmacological medications; there are a lot are environmental and supportive interventions that you can treat a baby’s experience and symptoms as well”.
Judgement	
Self-awareness of healthcare providers	“… there was, was a course they could take by enhanced services for women that actually helped you to examine your biases and … I think the best thing that you can do is if you can’t look after these women and these, and their babies without, you know, judgement, then you have to sort of say that and pass it on to somebody who does because I think you are doing so much harm”. “… that I actually wanted to quit…. it was … one that we were weaning down but it was still a withdrawal baby, and I was 1:1 with it because it was so bad, and I remember like wanting to leave and not come back the next day because it was just so hard, and I felt like I couldn’t take care of one baby, yeah. I just thought like I can’t do this, like I am not a good enough nurse to care for one baby … I think they are the hardest assignments”. “… like not every, every person is different, and it might not be a nurse with a real heart for people with mental health issues or you know how dependent are those people, and some who are actually afraid of it. You know because there is not really, so far, a training for it. You know, you have to have a heart for it, and you have to want to do this … you can’t learn compassion”.
Knowledge	
Desire for information	“I would love to see a mom come back who has had a good experience, like hey, I got a chance to, like community resources are out there for me, look how me and my baby are doing now and that is what I am thinking, like are people getting enough resources? Are we passing enough? Because honest to god, I don’t know any of the resources; I would love to know more resources I can pass on to the parents and say hey, there is help out there for you, not all hope is lost”.
Utilizing knowledge	“I think really educating caregivers, whoever that is, social workers, foster parents, bio[logical] families, particularly medical providers about trauma informed care … that there’s actually that American Academy of Pediatrics care tool kit; it’s a fully endorsed practice … and not just talking the talk but actually walking it and if you are walking it, then you will think about transportation, you will think about why they aren’t at the bedside … and instead of thinking, you know, that they’re doing something, maybe you will think they just don’t have the ability to get here today … And then you will think about offering them food when they come, instead of offering them judgement”.
Awareness of social circumstances	“… one of the biggest challenges is dealing certainly with the neonatal abstinence syndrome, but these are moms who don’t have a place to stay, they don’t have enough food, they, um, they don’t have the supports in place. All of the social determinants of health are not met”.

**Table 3 children-09-01517-t003:** Barriers to hope.

Primary Theme	Quotes
Mental Health	
Addiction	“… that’s been kind of the biggest thing for me, is the acknowledgement that this [maternal substance use] is something that is going on and you need to talk to patients about it and you need to understand it. I think understanding what resources are available and how to refer to those resources is also a big thing”.
Judgement	
Language	“… a NAS babe, well they are a baby, that’s first and foremost, they are a baby.”
Stigma	“I don’t think the stigma has changed enough. I think we are still approaching these families with the wrong heart at the bedside”.
Stereotypes	“I think that’s a, that’s a big thing we have to get through people’s heads is that drug use crosses all socioeconomic …”
Effects on relationships	“… and so, pregnancy is complicated because then all of a sudden it is not about them, the whole world is so focused on that fetus”.
Knowledge	
Fear	“… you add in these unknown factors and then I remember quite a few moms who I talked to say, you are going to put my baby on morphine? You are going to get my baby addicted? So, this is lack of knowledge that is very scary, let along seeing what their baby is going through. So, I think prenatal education is probably, might be the best bang for the buck in terms of, at least they are educated then. So, it would be less of surprise in terms of what they go through, and if their baby doesn’t need medical management, it’s kind of a bonus, right.”

## Data Availability

The data presented in this study are available on request from the corresponding author.

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
