# Peer review of "Fostering Hope: Comprehensive Accessible Mother-Infant Dyad Care for Neonatal Abstinence (CAIN)"

_children, 2022, doi:10.3390/children9101517_

Round 1
Reviewer 1 Report
Thank You very much for the study presented here. Although only a few family participants took part in this study, the reader gains some some insight in their worries - and the role care providers may play in overcoming these. Hope as a central topic -or as You worded it "the overarching theme"- for both "sides" is a feeling the reviewer has experienced quite often himself. Good luck for recruiting maternal/family participants in Your pan-Alberta project. Lookin forward to reading the results thereof.
As a non-native speaker I only observed very minute language issues (double spacing line 73, 190), table 1 "your co-workers" - perhaps editorial office could resolve these...
Reviewer 2 Report
Thank you for the opportunity to review the paper titled "Fostering hope: Comprehensive accessible mother-infant dyad care for neonatal abstinence (CAIN). Overall the paper is well written and contributes valuable insights to the body of knowledge on caring for infants with NAS and their families.
Table 1 contains a typographical error - "you co-workers" should be "your co-workers"
It is not clear why families were recruited to participate. Their participation raises the following questions. How is the interview guide in Table 1 relevant to family members? What questions were family members asked? How are the authors able to claim that no new themes emerged and data saturation was reached when only 2 members of the family group were interviewed? How were family members' data analysed and what were the specific themes from this data?
The limitations section mention challenges to recruitment of families yet somewhat dismiss this stating the focus of the work was on health care providers' experiences. It is not methodologically sound to state that the family perspective informed the identification and interpretation of themes.
The method of analysis is unclear as the authors initially state that content analysis was used but then mention thematic analysis. The method of analysis needs to be elaborated on and clarified.
Round 2
Reviewer 2 Report
Thank you for the opportunity to review this revised manuscript. The changes made by the authors have greatly enhanced the paper. I look forward to seeing this paper and future works the authors refer to related to families' perspectives published.